# The impact of global warming on the signature virulence gene, thermolabile hemolysin, of *Vibrio parahaemolyticus*

Weishan Zhang,[1,2] Keyu Chen,[1,2] Lin Zhang,[3] Ximeng Zhang,[4] Baoli Zhu,[1,2] Na Lv,[1] Kaixia Mi[1,2]

**ABSTRACT** Global warming is increasing human exposure to pathogens and has already had an impact on human health. *Vibrio parahaemolyticus*, a major pathogen causing foodborne illness, accumulates in numerous aquatic organisms and can be affected by environmental stressors such as increased replication errors or DNA damage resulting in point mutations. This could lead to an elevation in the mutation rate, which influences the expansion of *Vibrio* spp. habitats and the spread of associated diseases. In this study, a total of 241 strains were isolated from aquatic products imported or exported through China Customs between 2005 and 2010. The whole genomes of those strains were sequenced, revealing a highly significant level of genetic diversity. Our analysis identified 27 new sequence types (STs) ranging from ST2950 to ST2976. The global temperature trend since 1950 affects the thermolabile hemolysin gene (*tlh*) found in all *Vibrio parahaemolyticus* leading to mutant sites exhibiting similar trends as temperatures rise; seven high-frequency mutation hotspots (A180G, T552G, G657T, T858C, C1062T, A1137G, and T1179C) were identified along with two clinically specific sites (T259C and A951T) that may indicate adaptation over time due to climate change, leading to increased virulence potential for this bacterial species. These results provide insight into the genetics of *Vibrio parahaemolyticus* and provide a reference for subsequent research, identification, and monitoring efforts related to its spread.

**IMPORTANCE** In this study, *Vibrio parahaemolyticus* strains were collected from a large number of aquatic products globally and found that temperature has an impact on the virulence of these bacteria. As global temperatures rise, mutations in a gene marker called thermolabile hemolysin (*tlh*) also increase. This suggests that environmental isolates adapt to the warming environment and become more pathogenic. The findings can help in developing tools to analyze and monitor these bacteria as well as assess any link between climate change and vibrio-associated diseases, which could be used for forecasting outbreaks associated with them.

**KEYWORDS** *Vibrio parahaemolyticus*, global warming, thermolabile hemolysin (*tlh*), whole-genome sequencing, pathogenic potentials, mutation rate

C limate change has detrimental effects on human health, particularly through the exacerbation of human pathogenic diseases. A recent review showed that the majority of infectious diseases have been aggravated by climate change, suggesting that climate change is driving and reshaping the distribution of infectious diseases on a global scale (1). The world is experiencing severe challenges from global warming and a variety of severe climates, including extreme heat, heavy precipitation, ocean acidification, and sea level rise (2). The damage to the ecosystem and human health issues caused by global climate change may far exceed expectations (3, 4). A number of highly threatening disease outbreaks such as cholera, malaria, or dengue fever are strongly influenced by climate change. It is estimated that 34% of childhood illnesses and 36% of

Address correspondence to Kaixia Mi, mik@im.ac.cn, or Na Lv, lvna@im.ac.cn.

Weishan Zhang and Keyu Chen contributed equally to this article. The order of authors was determined based on their respective contributions to the research.

The authors declare no conflict of interest.

See the funding table on p. 15.

early childhood deaths worldwide are associated with changes in environmental factors (5). Smith et al. have hypothesized that the global warming mega-trend will result in at least 50,000 projected deaths from diarrhea and 60,000 deaths from malaria per year between 2030 and 2050 (6).

*Vibrio parahaemolyticus* (*V. parahaemolyticus*) was first isolated in Japan and caused a foodborne disease outbreak in 1950 (7). Due to its preference for a warm (>18℃), low-salinity (<25 ppt NaCl) habitat (8), climate change is thought to be associated with the transmission of *V. parahaemolyticus*. Outbreaks of vibrio-associated human illness linked to seawater temperature over the past decade have been accepted. During the hot summer months, up to 100% of oysters can be contaminated with *V. parahaemolyticus* (9, 10), and *V. parahaemolyticus* infection rates for human populations also tend to peak (11). Despite *V. parahaemolyticus* thriving in warmer waters, outbreaks of *V. parahaemolyticus* have been reported in colder regions, including Alaska, where rising temperature plays an important role in contributing to these outbreaks (12). Although previous studies have demonstrated a direct role of temperature in the pathogenicity of *V. parahaemolyticus* (13), little is known about the specific mechanisms involved in temperature-associated infections.

Therefore, there is an urgent need for analysis tools or monitoring markers to assess the causal link between macroecology and the spread of vibrio to forecast the outbreaks associated with these pathogens. Mastering the various molecular characteristics of *V. parahaemolyticus* and further exploring their relationship with temperature variation are important strategies to reduce infection and transmission (14). At present, studies conducted by collecting samples on a global scale are still relatively scarce, but it is necessary to characterize and study the molecular characteristics of *V. parahaemolyticus* on a global scale when the global economy and trade are becoming increasingly closely connected. Here, we isolated 241 strains of *V. parahaemolyticus* from a large volume of fish products imported and exported by China Customs between 2005 and 2010, which were further analyzed. Whole-genome sequencing (WGS) of these isolates elucidated their genomic profiles, described and analyzed their resistance and virulence profiles, and provided evidence that climate change led to specific mutations in the key virulence factor thermolabile hemolysin (*tlh*). Our findings can help improve the accuracy of quarantine of imported and exported seafood and suggest that using *tlh* as a key marker for detecting virulence can significantly reduce the labor and material resources required for detection. We provide a reference for subsequent research on the pathogenicity of *V. parahaemolyticus* and its prevention and control.

## RESULTS

### General genomic features and annotation of the sequenced *V. parahaemolyticus* isolates

A total of 294 *V. parahaemolyticus* strains were isolated from seafood imported and exported by China Customs, and these isolates were subjected to WGS and quality assessment. Except for 49 isolates with low sequencing quality, the pairwise average nucleotide identity (ANI) percentage showed that 241 isolates were highly similar (ANI > 95%) to the reference strain RIMD2210633 (Table S1) and could be used for further analysis. The total genome size of *V. parahaemolyticus* sequenced in this study was about 5.7 Mbp (4.7–6.5 Mbp); the average GC content was 45.1%; the average gene number was 4,815 (Table S2); the collection range spanned 5 continents, including 23 countries in Asia, Europe, North America, South America, and Oceania; the collection hosts contained shrimp, fish, crab, shellfish, and other 77 species of aquatic products (Fig. 1; Table S2); Prokka (v1.13.3) annotation (15) revealed approximately 90 tRNAs, 1 tmRNA, and 4 rRNAs; about 4,720 coding sequences (CDSs) were present, with an average of 3,121 functional proteins and 1,691 hypothetical proteins (Table S3). The distribution of clusters of orthologous groups (COGs) in the core, accessory, and unique genomes of the isolates showed that a larger proportion of the genomes were involved in [K] transcription and [L] replication, recombination, and repair (Fig. 2).

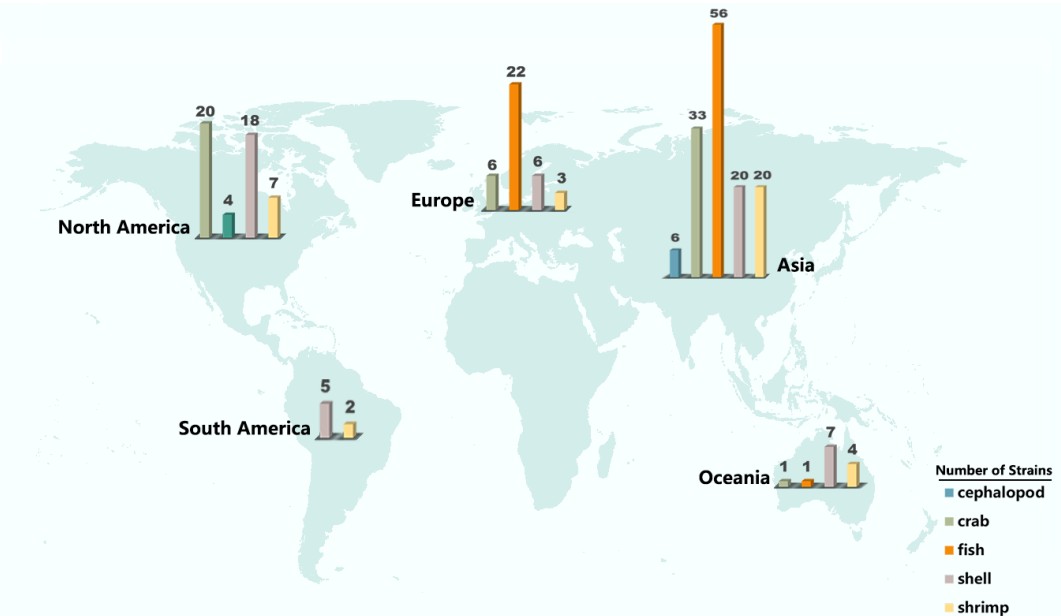

**FIG 1** Geographical distribution of *V. parahaemolyticus* strains collected by China Customs from 2005 to 2010. Square colors indicate the species carrying *V. parahaemolyticus*. The bar indicates the number of species shown in it. The world map was sourced from Natural Earth (https://www.naturalearthdata.com/).

## Multilocus sequence typing (MLST) analysis of 241 seafood isolates

Gonzalez-Escalona et al. described an MLST scheme for *V. parahaemolyticus* (16), which relies on seven housekeeping loci (*recA*, *dnaE*, *gyrB*, *dtdS*, *pntA*, *pyrC*, and *tnaA*) spanning both chromosomes of *V. parahaemolyticus*. This protocol was used to analyze the 241 isolates screened in this study. The corresponding MLST loci were extracted from the genomes of these isolates, and 99 untypable isolates (denoted as N/A) were excluded based on MLST typing from the PubMLST database (https://pubmlst.org/) because they had missing or unidentifiable alleles. The screened 143 isolates showed 80 STs were identified using the PubMLST database. Twenty-seven unique MLST allele profiles were identified, which were the newly identified STs in this study. Therefore, we submitted them to the PubMLST database, and the staff numbered the new STs (ST2950–ST2976) (Table S4).

## Clonal complex (CC) analysis

CC analysis was performed on the isolates using goeBURST (v1.2.1) (http://goeburst.phyloviz.net) (17). Overall, 107 STs were clustered into 103 single-locus variants and 4 double-locus variant doublet clusters (ST34–ST324, ST643–ST2960, ST411–ST2975, and ST1343–ST1295) (Fig. S1). The analysis of the strains collected in this study revealed a dispersed distribution without regional clustering.

## Comparative phylogenetic analysis of whole-genome sequencing-single nucleotide polymorphism (WGS-SNP)

We selected three reference strains because of their proven geographical representativeness (18) (VppAsia RIMD221063, VppUS2 FDA-R31, and VppX 10329 representing the Asian region, American region, and Pacific coast, respectively, denoted by "*") to establish a phylogenetic tree based on the WGS-SNP. Compared with the MLST-based tree, the WGS-SNP-based tree solves the problem of too few MLST loci to represent the whole genome. The parts that cannot be discerned in the MLST-based phylogenetic tree can be observed with higher classification resolution and accuracy of branching support values (19, 20). The tree was divided into four distinct clades (Fig. 3A and B), and further

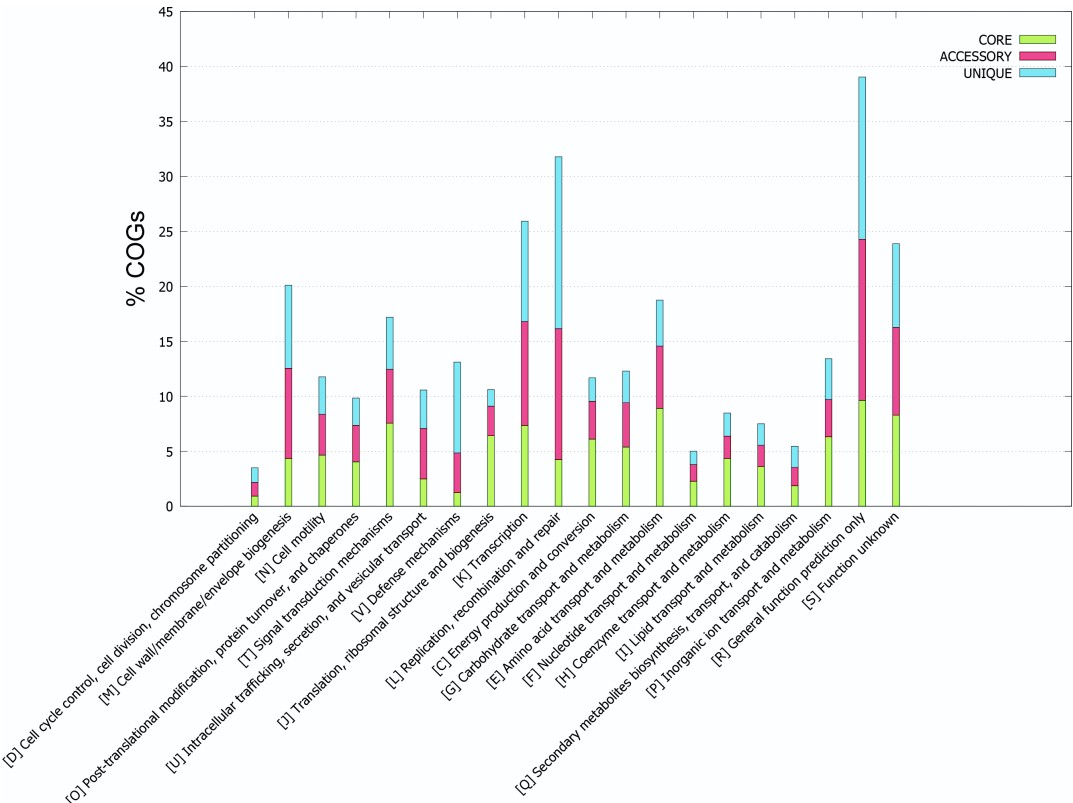

**FIG 2** Distribution of COGs in the core, accessory, and unique genomes of 294 *V. parahaemolyticus* strains using BPGA (v1.3). The research value is 0.75. The green bar represents the core genome. The pink bar represents accessory genomes. The blue bar presents unique genomes.

analysis showed that there was no direct relationship between phylogenetic clustering and STs, segregation sources, and hosts. The samples were evenly dispersed without any strong preferences.

## Antimicrobial resistance gene analysis of isolates

Antimicrobial drugs are also frequently used in the prevention and treatment of various diseases in aquaculture. However, with the widespread use of antimicrobial drugs, pathogens in aquatic products are developing resistance, rendering many known antimicrobial drugs ineffective (21). The genomic sequences of 241 isolates were screened for the presence and distribution of resistance genes and antimicrobial resistance gene families (AMR) using the Comprehensive Antibiotic Resistance Database (CARD) (22) (https://card.mcmaster.ca/). The results showed that the drug resistance genes of the isolates in this study mainly belonged to the efflux pump class (ABC antibiotic efflux pump, RND antibiotic efflux pump), β-lactamase resistance, and fluoroquinolone antibiotic resistance (Fig. 4A). The carbenicillin-hydrolyzing β-lactamase (CARB) gene family showed a strong correlation with the isolates in this study (*n* = 241). Seven CARB family genes were identified (CARB-17, CARB-18, CARB-19, CARB-20, CARB-21, CARB-22, and CARB-23). The most frequently detected gene was CARB-18 (*n* = 158), followed by CARB-22 (*n* = 48).

## Detection of virulence genes (*tdh*, *trh*, and *tlh*) of isolates

To further explore the virulence gene map of *V. parahaemolyticus*, we performed a BLAST search of the full virulence factors of pathogenic bacteria from the virulence factor database (VFDB) (http://www.mgc.ac.cn/VFs/main.htm) (Fig. 4B). Thermostable direct hemolysin (*tdh*) and/or *tdh*-related hemolysin (*trh*) genes, which play important roles in

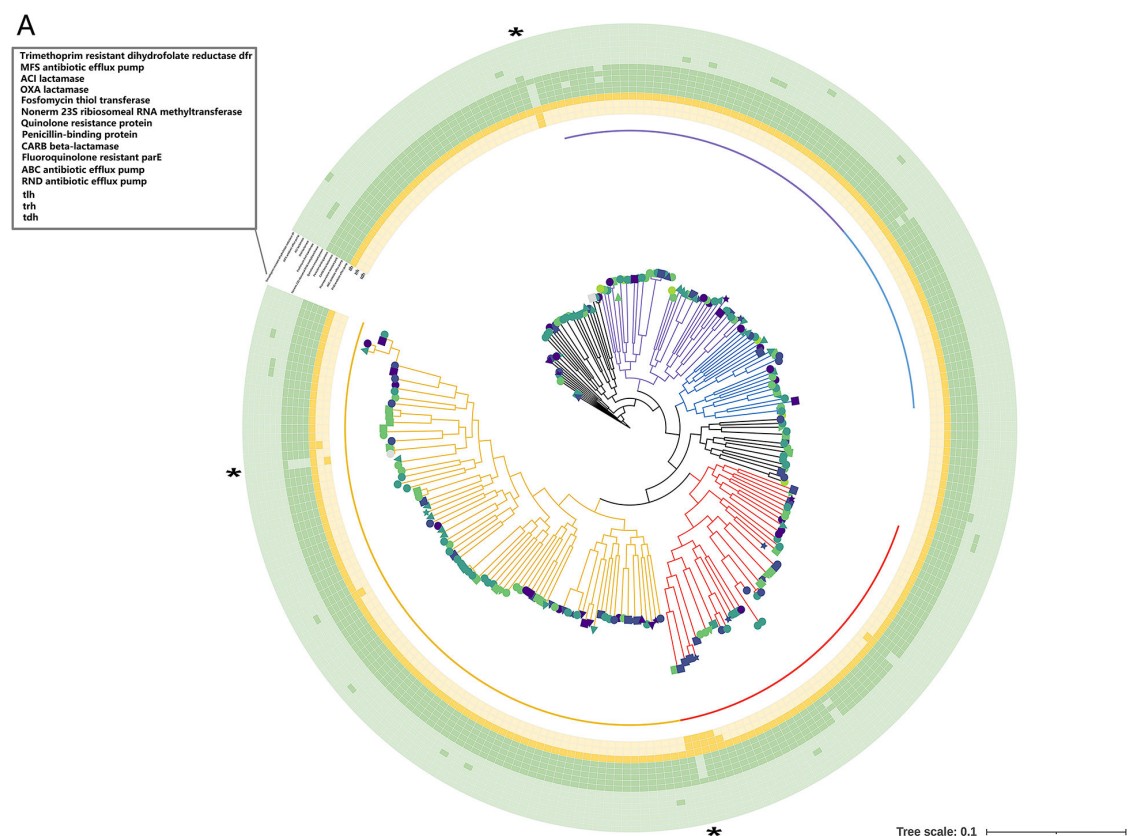

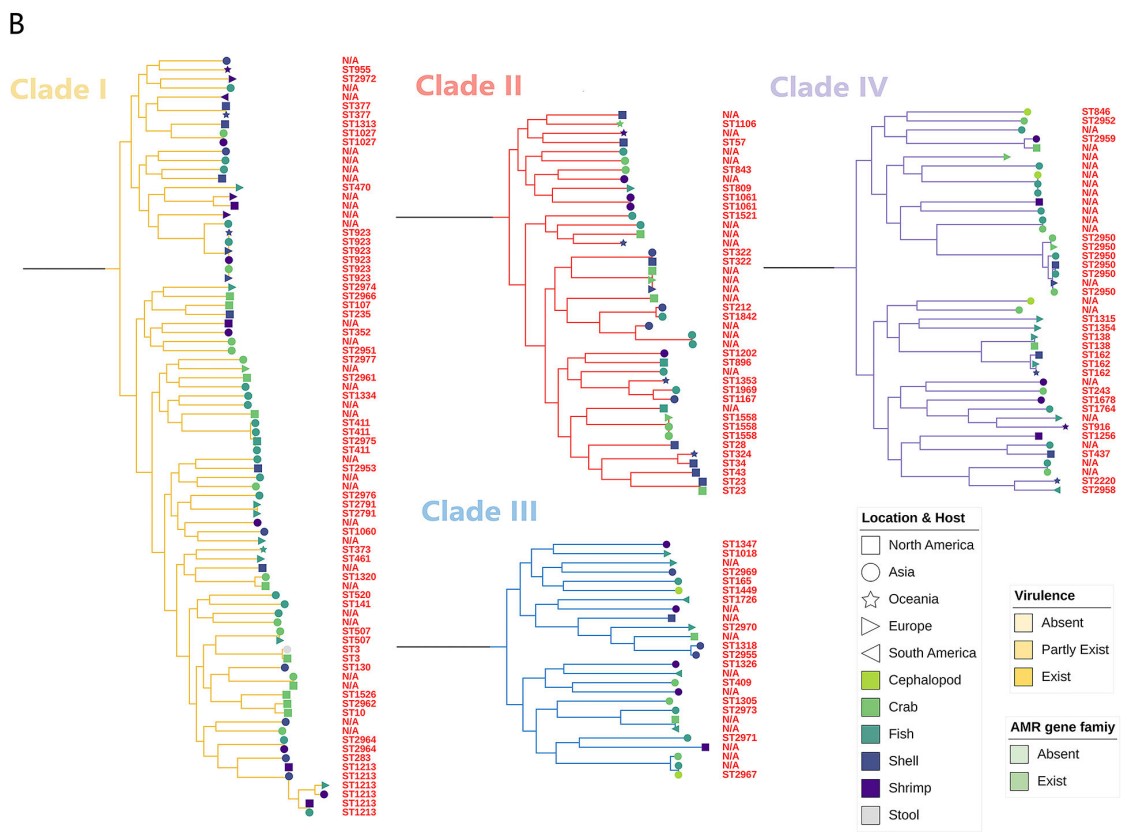

**FIG 3** (Continued)

**FIG 3** WGS-SNP-based phylogenetic tree of 241 *V. parahaemolyticus* strains. (A) *V. parahaemolyticus* strains were analyzed using KSNP (v3.1) (Kmer length = 20) and iqtree (v2.0.3); optimal model selection was performed using ModelFinder with 1,000 iterations. The reference strains (VppAsia RIMD2210633, VppUS2 FDA-R31, and VppX 10329) are marked with "*". (B) Branch colors represent the different populations that have been identified. Each color represents a specific population: yellow, clade I; red, clade II; blue, clade III; purple, clade IV.

the pathogenicity of *V. parahaemolyticus*, are considered good virulence predictors (23, 24). Of the 241 isolates, 7 carried *tdh* and/or *trh* (Fig. 5), vp.371 (*tdh+/trh+*), vp.384 (*tdh+/trh+*), vp.395 (*tdh+/trh+*), vp.332 (*tdh−/trh+*), vp.512 (*tdh−/trh+*), vp.650 (*tdh−/trh+*), and vp.528 (*tdh−/trh+*). These isolates, which were mainly from crabs and shells in Asia and North America, were mostly (71%) from clade II isolates (Fig. 3B). The characterization of

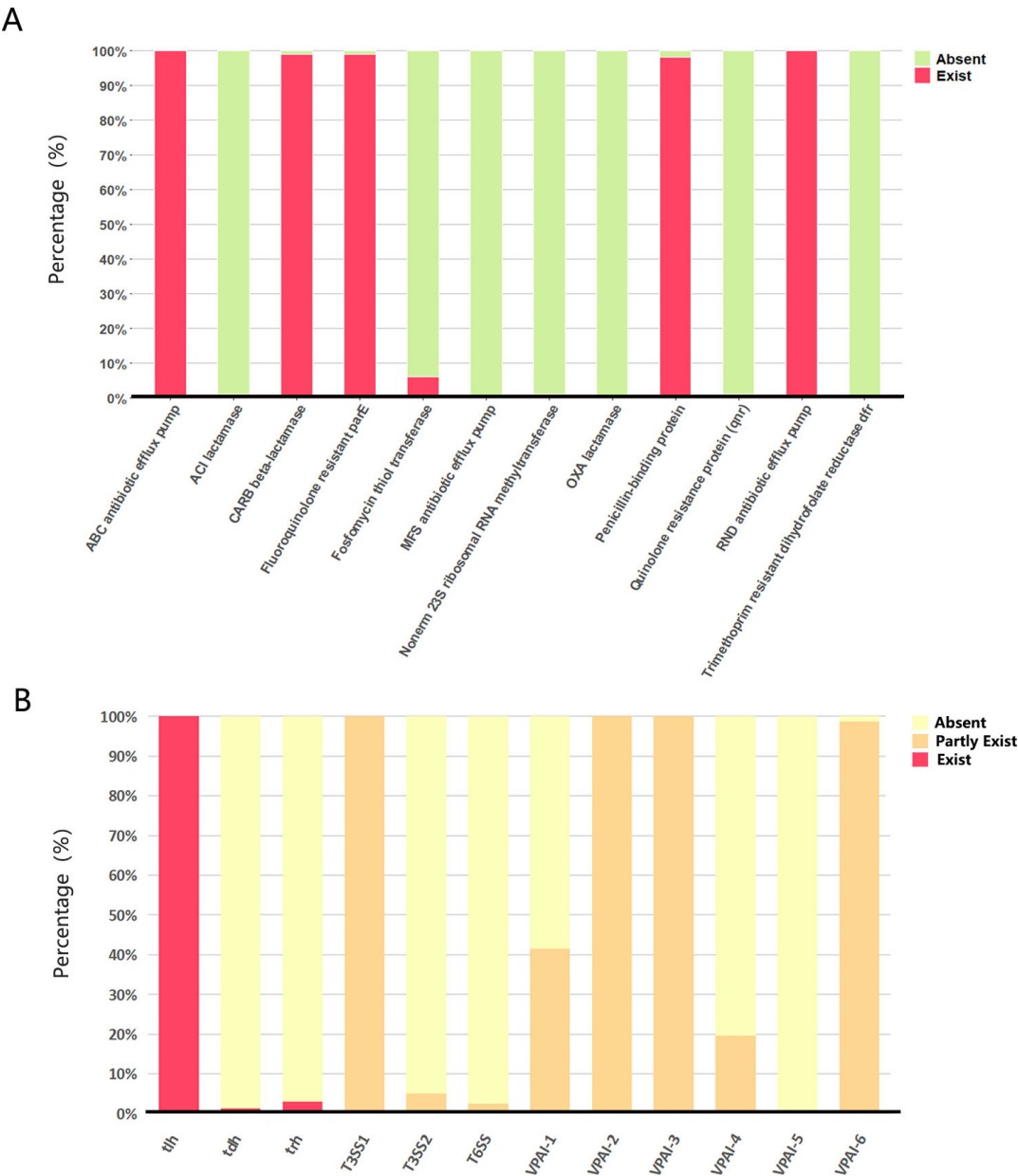

**FIG 4** Information on virulence factors and resistance genes of 241 *V. parahaemolyticus* strains. (A) Percentages of resistance genes and AMR genes in 241 *V. parahaemolyticus* strains. Red, exist; green, absent. (B) Distribution of virulence factors in *V. parahaemolyticus* strains. Yellow, absent; orange, partly exist; red, exist.

**A**

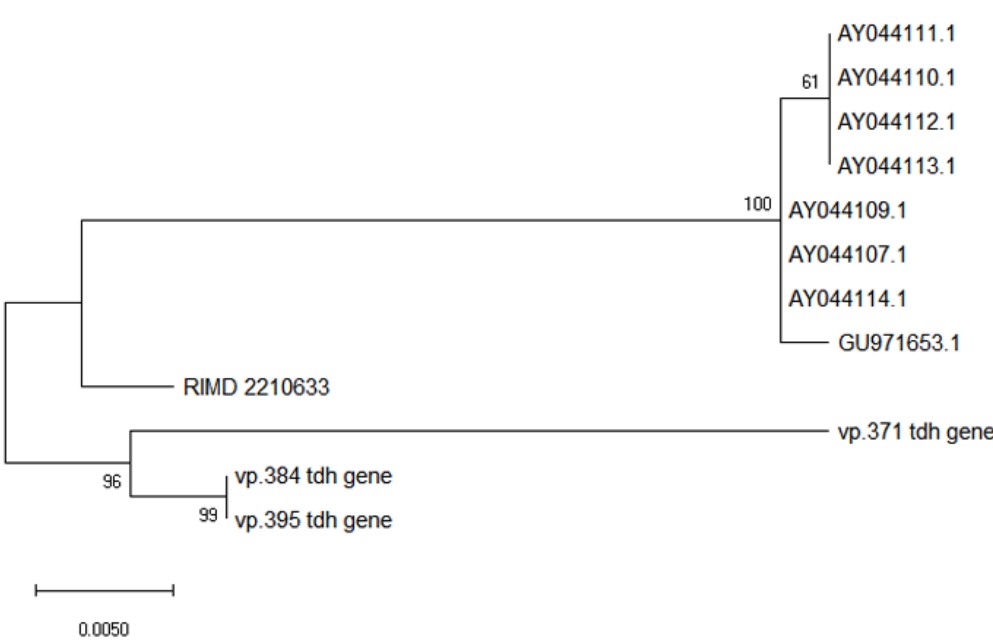

**B**

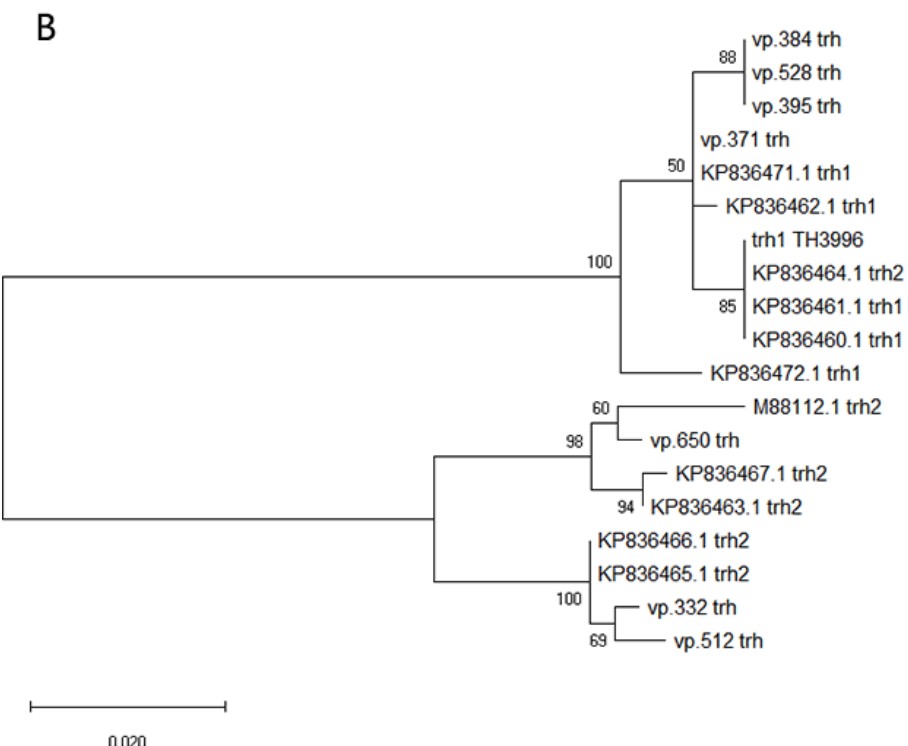

FIG 5   The presence of *tdh* (A) and *trh* (B) phylogenetic tree.

**TABLE 1** Ka/Ks ratios of *V. parahaemolyticus* contained *tdh* or *trh*[a]

| Strain | | TS | TV | NSM | SM | Ka/Ks |
|---|---|---|---|---|---|---|
| *tdh* | vp.371 | 2 | 1 | 1 | 3 | 2.00 |
| | vp.384 | 3 | 1 | 2 | 2 | 0.33 |
| | vp.395 | 3 | 1 | 2 | 2 | 0.33 |
| *trh* | vp.332 | 50 | 31 | 26 | 50 | 0.10 |
| | vp.371 | 2 | 1 | 1 | 3 | 0.10 |
| | vp.384 | 3 | 1 | 2 | 2 | 0.35 |
| | vp.395 | 3 | 1 | 2 | 2 | 0.35 |
| | vp.512 | 51 | 40 | 44 | 45 | 0.08 |
| | vp.528 | 3 | 1 | 2 | 2 | 0.10 |
| | vp.650 | 53 | 33 | 39 | 47 | 0.10 |

[a]NSM, non-synonymous mutations; SM, synonymous mutations; TS, transition; TV, transversion.

the isolates is presented in Table 1. Among them, the *trh* of vp.371, vp.384, vp.395, and vp.528 are *trh1*; vp.332, vp.512, and vp.650 are *trh2*.

In general, clinical isolates of *V. parahaemolyticus* can be classified as *tdh+/trh−*, *tdh+/trh+*, or *tdh−/trh+* strains based on their toxigenic activity (25). However, occasionally, strains without *tdh* or *trh* can also be isolated from clinics. They are pathogenic and have been shown to infect humans (24). Studies have also shown that environmental isolates of *V. parahaemolyticus* have virulence properties that make them potentially pathogenic to humans and aquatic animals (26). Thus, there must be other active virulence factors in these isolates. In fact, all *V. parahaemolyticus* strains possess an additional toxin, thermolabile hemolysin (*tlh*), which is also a specific marker widely used for *V. parahaemolyticus* detection (27, 28). All isolates collected in this study also contained *tlh* (*n* = 241) (Fig. 4B).

## Impact of environmental climate on the pathogenic potential

Studies have already shown that the global warming trend will have a strong impact on the micro-ecological balance, and such impacts may even alter the vectors of disease pathogens, as well as their survival and mutation (29). The warming of the oceans, the changes in plankton populations, and the increase in human activities in coastal waters could substantially change the exposure of *V. parahaemolyticus* to humans, thereby increasing the overall risk of human exposure to infection (30–32). Thus, global warming is bound to have a significant impact on pathogen concentrations in water bodies and even aquatic organisms (33). To better explore whether there is a relationship between temperature and *V. parahaemolyticus*, we first collected information on all isolates from 2005 to 2010, which could be pinpointed to the year and month of collection. They were grouped by source (clinic, 778 strains; environment, 739 strains; this study, 241 strains). Then, we compared the corresponding global mean temperatures (http://berkeleyearth.org/data/) and found that *V. parahaemolyticus* detection was mainly concentrated in July–August at high temperatures (Fig. 6A). At the same time, as the time span lengthened, we found that the number of single nucleotide polymorphisms (SNPs) in *tlh* also showed an increasing trend in the annual increase in global temperature (Fig. 6B; Fig. S2), suggesting that global warming may be inextricably linked to the above-mentioned mutations in *tlh*. Moreover, *tlh* may be a breakthrough in successfully explaining why environmental isolates of *tdh−/trh−* may also show pathogenic potential.

## Identification of *tlh* mutations

We performed a comprehensive analysis of all *tlh* full-length sequences (1,257 bp), including 778 clinical strains, 739 environmental strains, and 241 strains collected in our study. These sequences were compared with the *tlh* gene of the reference strain (RIMD2210633). Through this analysis, we identified a total of 48 SNPs. We calculated the frequency of mutations at each site and determined sites with a mutation frequency

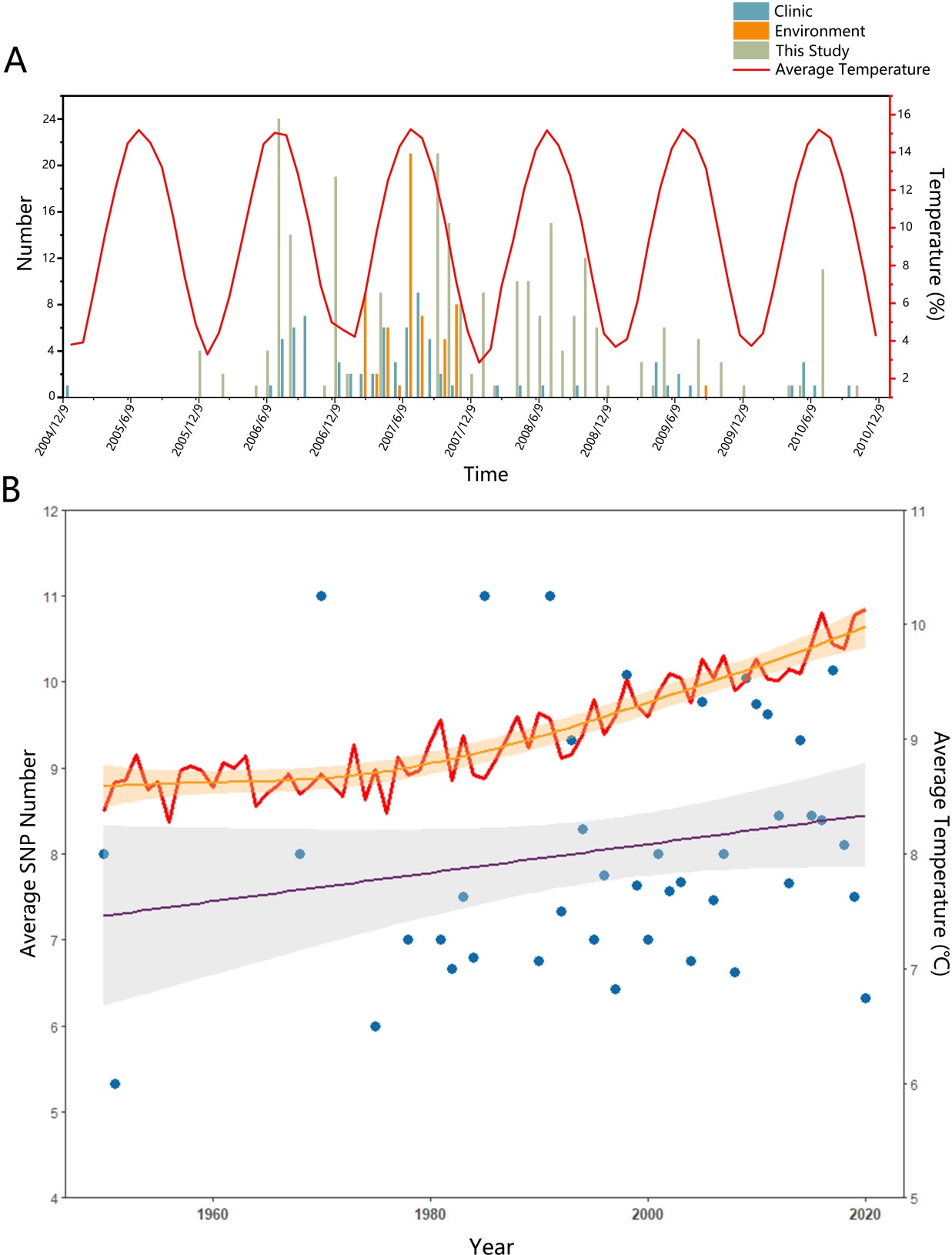

**FIG 6** The relationship between temperature and *V. parahaemolyticus*. (A) The collection date of *V. parahaemolyticus* isolated by different sources (clinic, environment, this study) and the average temperature information from 2005 to 2010. Cyan, clinic; orange, environment; dark green, isolated strains collected in

**FIG 6** (Continued)

this study. The red line indicates the global monthly average temperature. (B) The *tlh* was progressively more mutated with global temperatures increased. The red line indicated the global average temperature and the orange line was its fitting curve. The blue point indicated the average SNP number of *tlh* in different years, and the purple line was its fitting curve. The light orange and gray shadows indicated the 99% confidence interval.

higher than 50% as high-frequency mutation points. We found seven high-frequency mutation hotspots: A180G, T552G, G657T, T858C, C1062T, A1137G, and T1179C (Fig. 7). Additionally, we identified two specific sites, T259C and A951T, which were exclusively present in the clinical isolate (highlighted in red in Fig. 7). These mutations exhibited a high frequency of occurrence exceeding 15% among clinical strains. To further analyze the potential significance of these mutations, we calculated the protein-protein binding probability for the identified mutation sites (Table S5). Our analysis revealed substantial variation in the ability of different sites to engage in protein-protein interactions, with probabilities ranging from 0.8% to 59.3% (Table S5). Notably, two mutation sites were predicted to be associated with a high probability of protein-protein binding sites (Table S5). These findings suggest a potential link with these mutations and pathogenicity through protein-protein interactions. However, experimental validation is necessary to confirm these predictions. After conducting protein structure simulations of TLH, we

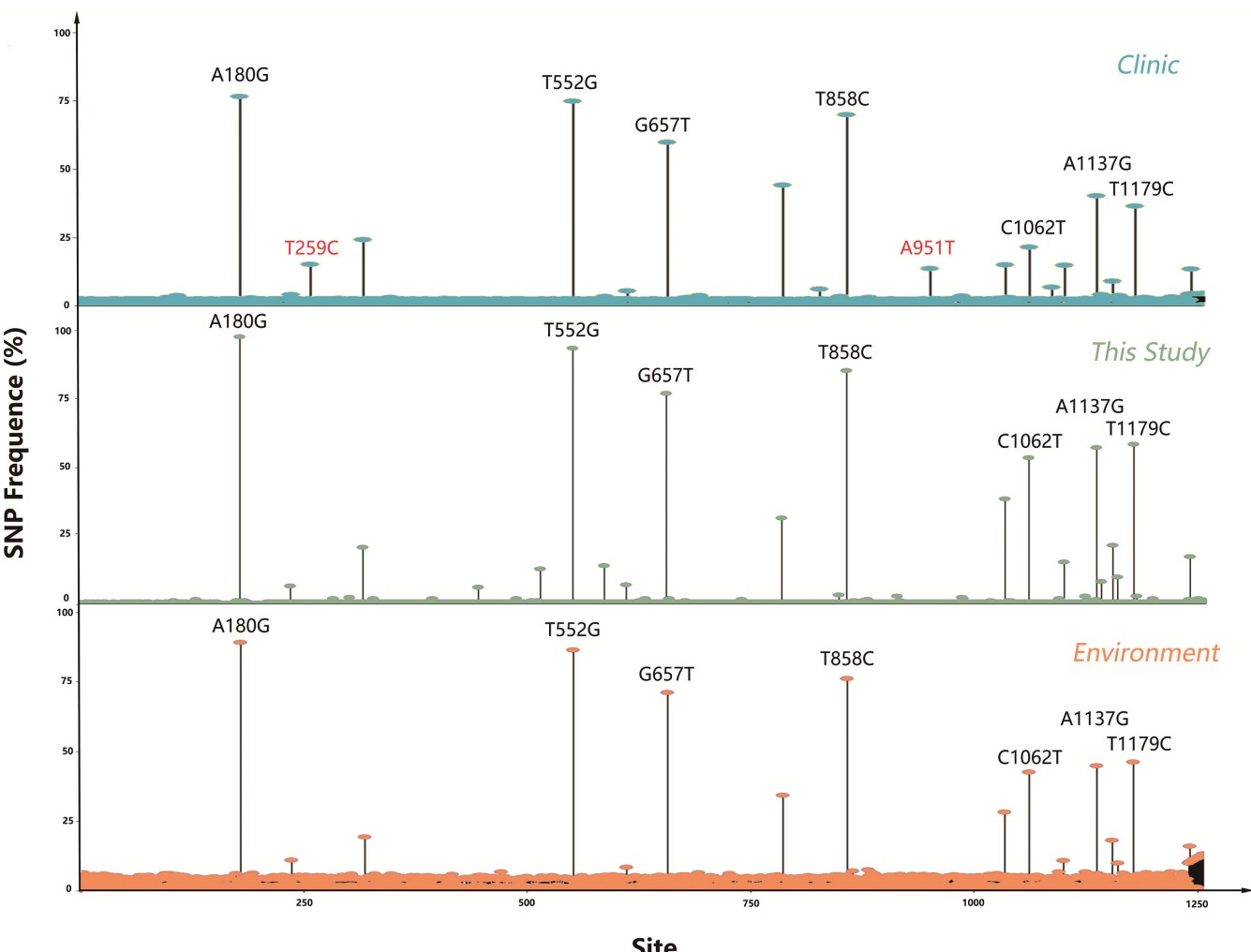

**FIG 7** The distribution of *tlh* mutations. Seven high-frequency mutation hotspots were noted: A180G, T552G, G657T, T858C, C1062T, A1137G, and T1179C, and two unique (T259C, A951T) mutation spots of the clinic were noted by red color. The x-axis indicates the site in *tlh*, and the y-axis indicates the SNP frequency. Cyan, clinic; orange, environment; dark green, isolated strains collected in this study.

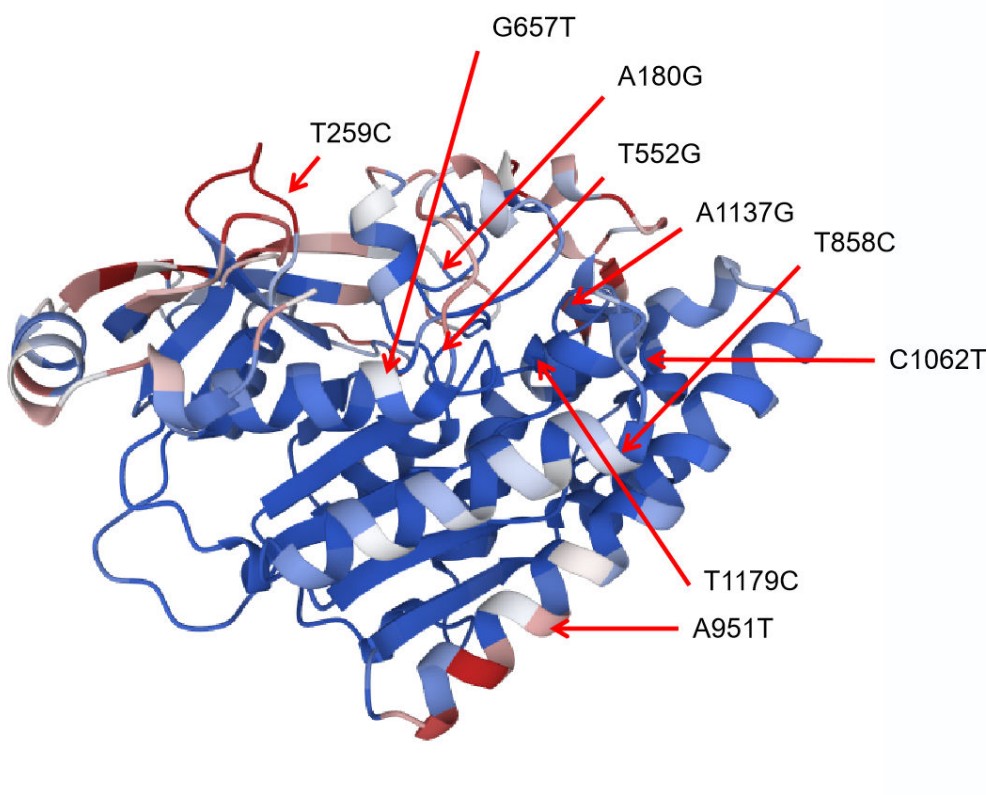

**FIG 8** The structure prediction and probabilistic analysis of protein-protein binding sites of TLH. The red arrows point to the important SNP sites identified in this study.

identified and compiled a list of nine mutation sites that show a correlation with protein changes (Fig. 8). Notably, the T259C mutation leads to the substitution of the amino acid Phe with Leu at this specific position, which residues within the loop region of the N-terminal domain. Furthermore, the G657T mutation induces a protein alteration from Glu to Asp at a position within the characteristic GDSL motif, which is associated with the esterase-lipases family of TLH esterase-lipases (Table S5) (34, 35).

To further determine whether *tlh* hypermutation sites affected other virulence genes, we analyzed their correlation and visualized them. The results showed that T259C and G657T were significantly negatively correlated with the components of the type III secretion system 2 (T3SS2) ($P < 0.05$), and the unique site T259C in the clinical isolates showed a more significant negative correlation with each component of the T3SS2 ($P < 0.001$) (Fig. 9). As above, we found *tlh* is likely to be an important marker of the pathogenic potential of environmental isolates.

## DISCUSSION

Climate change is increasing the temperature of the world's waters, which in turn is increasing the spread of waterborne diseases caused by bacterial pathogens, such as vibrios (36). Vibrios, which can cause diseases in both humans and aquatic animals, are bacteria that prefer warm, low-salinity environments (37–39). Vibrio infections have become more common in humans as ocean temperatures continue to rise owing to

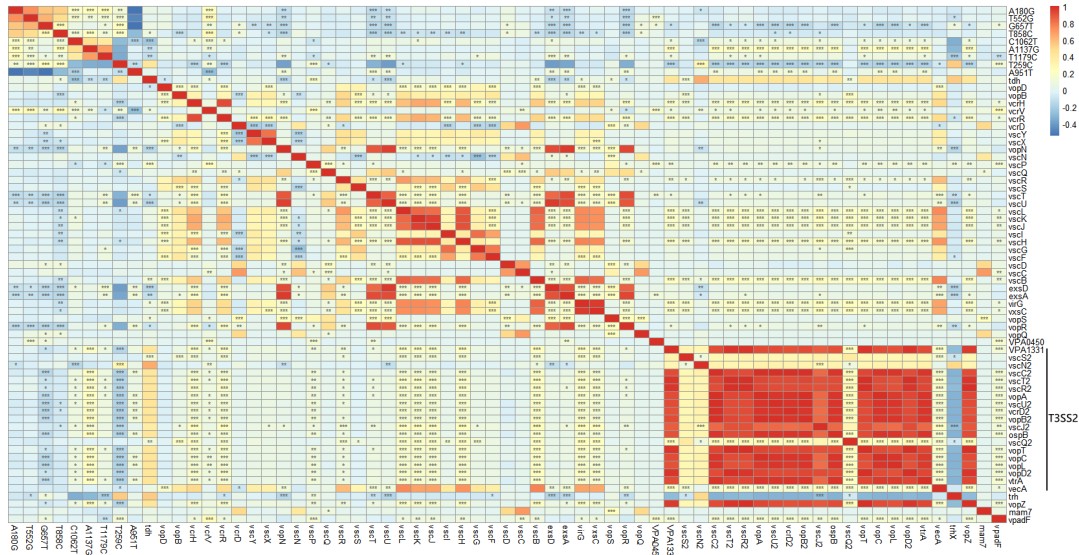

**FIG 9** The correlation analysis between *tlh* SNPs that we detected and other virulence genes. Red indicates a positive correlation, blue indicates a negative correlation, and the color shade represents the strength of the correlation."***" means the false discovery rate (fdr) value is less than 0.001, "**" means the fdr value is greater than 0.001 and less than 0.01, and "*" means the fdr value is greater than 0.01 and less than 0.05.

climate change. Therefore, understanding how climate change affects marine ecosystems would make these bacteria useful indicators. The combination of microbiological, genomic, epidemiological, climatic, and oceanographic data provides a better understanding of emerging waterborne diseases (40, 41).

The understanding of *V. parahaemolyticus* has been limited by focusing only on certain geographical areas rather than looking at a global scale (42–44). Studies have shown that human activity has increased the genetic mixing of *V. parahaemolyticus* populations worldwide (18). In this study, 241 diverse *V. parahaemolyticus* isolates were collected (Fig. 1). Using high-resolution whole-genome sequencing, we substantially complemented the database of *V. parahaemolyticus* and provided more detailed information for further studies (Table S2). We then analyzed the sequences of 143 strains and found 27 new STs (Table S4) and performed COG analysis, which revealed frequent mutations and recombination events in environmental isolates (Fig. 2). Due to the high genetic diversity of the *V. parahaemolyticus* genome, we found no significant correlation between different STs, hosts, and geography (Fig. 3; Fig. S1). The study also found that most environmental isolates with signature virulence factors (*tdh* and *trh*) clustered in clade II (Fig. 3B), suggesting that these isolates may evolve into virulent pathogens.

The *tdh* and *trh* genes are two virulence factors that are associated with the hemolysis of *V. parahaemolyticus* and with the cytotoxic activity of host cells (45). Consistent with the results of Bilung et al. (46), Lopatek et al. (47), and Ottaviani et al. (48), the detection rate of *tdh* (1.2%) was lower than *trh* (2.9%) in this study. We examined the genetic variations and changes in amino acid substitutions in these seven samples carrying *tdh* and/or *trh* (Fig. 5). Our results suggest that purifying selection (Ka/Ks < 1) is more common in *trh* than in *tdh* (Table 1). Further research is needed to determine the reasons for the low prevalence of *tdh* and *trh* in the environment and food, but the high prevalence of foodborne *V. parahaemolyticus* infections. Similarly, many *tdh*− and/or *trh*− clinical strains have been identified (45), but virulence factors have not been detected in clinical isolates. *tdh* and *trh* are not entirely reliable as the gold standard for determining whether an isolate is pathogenic (49). Through comparative genomic analysis of clinical strains, Ronholm et al. (50) suggested several other notable virulence markers, including the potential virulence factor *tlh* (47, 51). The function, pathogenesis, and gene epidemic of *tlh* are still unknown. We examined all available full-length *tlh* and found several specific locations (A180G, T552G, G657T, T858C, C1062T, A1137G, and

T1179C for isolated environmental *V. parahaemolyticus* strains and T259C and A951T for isolated clinical strains) where mutations occurred frequently (Fig. 7). In addition, as predicted by ScanNet (http://bioinfo3d.cs.tau.ac.il/ScanNet/), T259C and A951T may be particularly important for bacterial binding to the receptor, since their protein-protein binding can be up to about 50%. Meanwhile, mutations in *tlh* (T259C and G657T) were found to negatively affect T3SS2, which is an important virulence factor responsible for the secretion of toxins into human cells and plays a major role in intestinal diseases (52–54). The study also found a correlation between the T259C mutation and deletion in the T3SS2 (Fig. 9), suggesting that T259C may compensate for the loss of T3SS2 in terms of host enterotoxicity. Further experiments are required to explore the effect of T259C on *tlh*.

The world is becoming warmer, causing severe changes in the environment, particularly in oceans. One consequence of this is that certain diseases can worsen owing to climate change (55). Studies have shown that *Vibrio* spp. spread as sea surface temperatures rise (56), and they adapt to the environment by using cellular, genetic, and physical changes to resist stress and maintain normal function (55, 57, 58). Consistent with previous studies, we showed that mutations occur in *tlh* and *tlh* was progressively more mutated with increasing global temperature (Fig. 6B; Fig. S2), which may be potentially harmful to humans. Therefore, it is important to monitor and study *tlh* to understand the potential dangers posed by *V. parahaemolyticus* due to climate change. Our results greatly complement the information of *V. parahaemolyticus* and are beneficial to the subsequent research, identification, and surveillance related to its transmission.

## MATERIALS AND METHODS

### Isolation and identification of bacterial strains

A total of 294 strains of *V. parahaemolyticus* were isolated from various types of aquatic products collected from customs according to the procedure of the National Standard of the People's Republic of China (GB 4789.1–2016). Samples (25 g) were placed in 225 mL buffered peptone water (Land Bridge Technology Co., Ltd., Beijing, China) at 37°C and incubated for 8 h. The pre-enriched culture (1 mL) was transferred to 10 mL Thiosulfate Citrate Bile Salts Sucrose Agar (TCBS, Land Bridge Ltd., Beijing, China). After incubation in TCBS for 18 h at 36°C, one lap of each broth was added to 3% sodium chloride tryptone soy agar plate (Land Bridge Ltd., Beijing, China) incubated at 36°C for 18–24 h. After incubation, the plates were screened and identified for morphology typical of *V. parahaemolyticus* colonies, and DNA sequencing was performed.

### DNA sequencing, assembly, and annotation

We used the QIAamp DNA Mini Kit to extract genomic DNA from isolates and perform WGS (Qiagen, Hilden, Germany). A DNA library was constructed using a NEXTflex rapid DNA-seq kit (Illumina, San Diego, CA, USA) according to the manufacturer's instructions. Briefly, a paired-end library with an insert size of 350 bp for each sample was constructed and sequenced with a 150-bp read length from each end on the Illumina HiSeq 2500 platform (Illumina, San Diego, CA, USA). Trimmomatic was used to intercept adapters from sequencing reads generated by Illumina Hiseq. The reads were assembled using the default parameters of Unicycler (v0.4.7), assembled using SPAdes (v3.13.0) for short reads, and corrected using Pilon (51, 59, 60). We used Prokka (v1.13.3) to annotate the genome and Prodigal (v2.6.3) to identify open-reading frames.

### ANI and MLST analysis

Of the 294 isolates, 49 were sequenced with low quality, and the genomes of the remaining 245 isolates and the standard strain RIMD2210633 were subjected to average nucleotide identity ANI analysis using FastANI (v1.33) (61), with 95% as the threshold. The

genomic data of the 241 isolates screened were then compared with the *V. parahaemolyticus* typing database in PubMLST (https://pubmlst.org/) (and the housekeeping genes *dnaE*, *gyrB*, *recA*, *dtdS*, *pntA*, *pyrC*, and *tnaA*), and the new ST data identified in this experiment were uploaded to PubMLST to obtain the numbers (ST2950–ST2976).

## Core genome and COG analysis

The genomic data of 241 *V. parahaemolyticus* isolates were analyzed by core genome analysis and COG analysis using BPGA (v1.3) (62), with a research value of 0.75.

## Assignment to clonal complexes

To gain insight into the clonal relationships among the analyzed *V. parahaemolyticus* isolates, we conducted an analysis of their STs using goeBURST (v1.2.1) available at http://goeburst.phyloviz.net (17, 63). The analysis focused on identifying isolates belonging to the same clonal complex, which was determined by the presence of a minimum of five out of seven identical STs.

## WGS-SNP phylogenetic analysis

Whole-genome sequences were obtained from 241 *V. parahaemolyticus* strains (Table S2), along with 3 additional strains (VppAsia RIMD221063, VppUS2 FDA-R31, and VppX 10329 representing the Asian region, American region, and Pacific coast, respectively) (18). WGS-SNP analysis was performed using KSNP (v3.1) with a kmer length of 20 to analyze these sequences (64). The resulting data were utilized to construct maximum likelihood phylogenetic trees using iqtree (v2.0.3) (65). ModelFinder was employed with 1,000 iterations to perform optimal model selection for this analysis. Phylogenetic tree results were subsequently embellished using iTOL (https://itol.embl.de/).

## Statistics on the distribution of drug resistance genes

The genomes of *V. parahaemolyticus* isolates were compared with the CARD (https://card.mcmaster.ca/), and the distribution of resistance genes and AMR was counted after comparison.

## Virulence gene and gene island distribution analysis of isolated strains

The VFDB (http://www.mgc.ac.cn/VFs/main.htm) database was used for virulence gene annotation. Gene island data were obtained from a previous study of the RIMD2210633 gene islands (VAPI1–VAPI6). Subsequently, the gene island data of 241 *V. parahaemolyticus* isolates were analyzed using BLAST to determine their distribution.

## SNP analysis of virulence gene

The SNP of the important virulence genes *tdh* and *tlh* of 241 *V. parahaemolyticus* isolates were analyzed by Bioaider using RIMD2210633 as the reference strain (66); the SNP of the *trh* gene were analyzed using TH3996 as the reference strain (67). For the analysis of SNPs in *tlh*, the SNPs were ranked in order of frequency of SNPs from high to low, and the 48 more authentically present SNPs were finally selected after removing random effects. Ka/Ks calculations for *tdh* and *trh* were performed using DNAman.

## Correlation analysis between *tlh* SNPs and virulence genes

The Hmisc package in R was used for correlation analysis of virulence genes and *tlh* SNP of *V. parahaemolyticus* isolates. The results were visualized using the ggplot2 package. "***" means the fdr value is less than 0.001, "**" means the fdr value is greater than 0.001 and less than 0.01, and "*" means the fdr value is greater than 0.01 and less than 0.05.

## Structure prediction and probabilistic analysis of protein-protein binding sites

The prediction of TLH structure was completed using SWISS-MODEL (https://swiss-model.expasy.org/). Protein-protein binding site prediction was performed by using ScanNet (http://bioinfo3d.cs.tau.ac.il/ScanNet/) (68).

## Associations between temperature and *V. parahaemolyticus* strains

*V. parahaemolyticus* isolates from 241 isolates as well as from the NCBI database were plotted against global monthly average temperature data collected from Berkeley Earth (http://berkeleyearth.org/data/). The number of SNPs in *tlh* was counted, and the mean number of SNPs per year in *tlh* was calculated. The average temperature per year was counted and plotted using ggplot2, where the trend lines for the average annual temperature and average number of SNPs per year *tlh* were fitted using curve fitting and generalized linear equations, respectively.

## ACKNOWLEDGMENTS

This work was supported by grants from the National Key Research and Development Program of China (2022YFC2303200 to K.M.), the National Natural Science Foundation of China (31970136 and 32170181 to K.M.), International Joint Research Project of the Institute of Medical Science, University of Tokyo (Extension-2019-K3006 to K.M.), and the Open Project Program of CAS Key Laboratory of Pathogenic Microbiology and Immunology (CASPMI202201 to K.M.).

K.M., N.L., and B.Z. conceived and designed the experiments; L.Z., X.Z., W.Z., and K.C. collected the samples and performed the experiments; W.Z. and K.C. analyzed the data; W.Z., K.C., and K.M. wrote the paper; K.M., N.L., and B.Z. did the editing and proofreading. All authors have read and agreed to the published version of the manuscript.

## AUTHOR AFFILIATIONS

[1]CAS Key Laboratory of Pathogen Microbiology and Immunology, Institute of Microbiology, Chinese Academy of Sciences, Beijing, China
[2]Savaid Medical School, University of Chinese Academy of Sciences, Beijing, China
[3]Shijiazhuang Customs Technology Center, Hebei, China
[4]Science and Technology Research Center of China Customs, Beijing, China

## AUTHOR ORCIDs

Weishan Zhang  http://orcid.org/0000-0001-5511-6631
Keyu Chen  http://orcid.org/0009-0003-9821-1964
Baoli Zhu  http://orcid.org/0000-0001-5326-9503
Na Lv  http://orcid.org/0000-0003-0603-6170
Kaixia Mi  http://orcid.org/0000-0002-4399-2930

## FUNDING

| Funder | Grant(s) | Author(s) |
| --- | --- | --- |
| MOST \| National Key Research and Development Program of China (NKPs) | 2022YFC2303200 | Kaixia Mi |
| MOST \| National Natural Science Foundation of China (NSFC) | 31970136 | Kaixia Mi |
| MOST \| National Natural Science Foundation of China (NSFC) | 32170181 | Kaixia Mi |
| International Joint Research Project of the Institute of Medical Science, University of Tokyo | Extension-2019- K3006 | Kaixia Mi |

| Funder | Grant(s) | Author(s) |
| --- | --- | --- |
| Open Project Program of CAS Key Laboratory of Pathogenic Microbiology and Immunology | CASPMI202201 | Kaixia Mi |

## AUTHOR CONTRIBUTIONS

Weishan Zhang, Conceptualization, Data curation, Formal analysis, Methodology, Validation, Visualization, Writing – original draft, Writing – review and editing | Keyu Chen, Conceptualization, Data curation, Formal analysis, Methodology, Software, Validation, Visualization, Writing – original draft, Writing – review and editing | Lin Zhang, Resources | Ximeng Zhang, Resources | Baoli Zhu, Conceptualization, Data curation, Methodology, Project administration, Resources, Supervision, Validation, Visualization, Writing – review and editing | Na Lv, Conceptualization, Data curation, Methodology, Project administration, Resources, Supervision, Validation, Visualization, Writing – review and editing | Kaixia Mi, Conceptualization, Data curation, Formal analysis, Funding acquisition, Investigation, Methodology, Project administration, Supervision, Validation, Visualization, Writing – original draft, Writing – review and editing

## DATA AVAILABILITY

Data of *V. parahaemolyticus* strains were deposited in China National Microbiology Data Center (NMDC) with accession numbers NMDC60033607–NMDC60033900. The source data underlying Fig. 4, 6, 7, 9, and S2 are provided in the supplemental material.

## ETHICS APPROVAL

This article complies with ethical standards for research.

## ADDITIONAL FILES

The following material is available online.

### Supplemental Material

**Supplemental source data for Fig. 4 (Spectrum01502-23- S0001.xls).** (A) Distribution of drug resistance genes, and annotation data. (B) Distribution of virulence factor genes, and annotation data.
**Supplemental source data for Fig. 9 (Spectrum01502-23- S0002.xls).** Correlation analysis between *tlh* SNPs and virulence genes.
**Supplemental figures and tables (Spectrum01502-23- S0003.pdf).** Fig. S1 and S2; Tables S1 to S5.
**Supplemental source data for Fig. 6, 7, and S2 (Spectrum01502-23- S0004.xls).** Global temperatures and SNP analysis.

### Open Peer Review

**PEER REVIEW HISTORY An accounting of the reviewer comments and feedback. (review-history.pdf).**

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
