## [Reviewer comments · Microbiology Spectrum]

Microbiology Spectrum

The impact of global warming on the signature virulence gene thermolabile hemolysin gene of *Vibrio parahaemolyticus*

Weishan Zhang, Keyu Chen, Lin Zhang, Ximeng Zhang, Baoli Zhu, Na Lv, and Kaixia Mi

Corresponding Author(s): Kaixia Mi, Institute of Microbiology, Chinese Academy of Sciences

Review Timeline:

Submission Date:	April 10, 2023
Editorial Decision:	July 22, 2023
Revision Received:	August 17, 2023
Accepted:	September 5, 2023

Editor: Kevin Theis

Reviewer(s): Disclosure of reviewer identity is with reference to reviewer comments included in decision letter(s). The following individuals involved in review of your submission have agreed to reveal their identity: Isabella Cubillejo (Reviewer #1)

Transaction Report:

DOI: <https://doi.org/10.1128/spectrum.01502-23>

July 22, 2023

Dr. Kaixia Mi
Institute of Microbiology, Chinese Academy of Sciences
NO.1 Beichen West Road, Chaoyang District,
Beijing
China

Re: Spectrum01502-23 (The impact of global warming on the signature virulence gene thermolabile hemolysin gene of *Vibrio parahaemolyticus*)

Dear Dr. Kaixia Mi:

Thank you for submitting your manuscript to Microbiology Spectrum. Two reviewers have provided feedback on the manuscript. You will see from their comments below that both view the manuscript and study favorably. However, some modifications are needed. Specifically, see the comments regarding improving the presentation of figures and tables, and the comments suggesting further grammatical editing. If you have any questions, please let mw know.

Link Not Available

Sincerely,

Kevin Theis

Journals Department
Reviewer comments:

Reviewer #1 (Comments for the Author):

The methods of the paper are robust, and the tie-in to global warming is prescient. Minor misspelling or grammar corrections are required, as well as clarifications on methods or results.

Line 68: Change "aggregated" to "aggravated".

Line 89: Clarify why risk of infection is indirect?

Line 197: Remove italics from "strains"

Line 226-228: Clarify if other unlabeled high-frequency mutation hotspots in Figure 10 not chosen for further analysis were initially considered, but had a protein-protein binding probability of less than 50%.

Line 324: We used, or, Trimmomatic was used

Line 327: We used Prokka, or, Prokka was used.

Line 295: Remove space between Fig, to stay consistent in the paper

Line 673: Typo, change to Figure Legend.

S1: Does this analysis take an increase in surveillance or strain collection into account?

Reviewer #2 (Comments for the Author):

The manuscript aims to provide further genomic understanding of *Vibrio parahaemolyticus* and to specifically develop understanding of diversity in its virulence factors and how the mutation potential of some of the key factors correlates with ambient temperatures.

The general approach of the study was to sequence the genomes of 241 strains that were isolated from aquatic products imported or exported from China Customs between 2005-2010. Specific emphasis was placed on variation in the thermolabile hemolysin gene (*tlh*) gene in the context of temperature.

The conclusion of the study was that the *tlh* gene is likely to be an important marker of the pathogenic potential of environmental isolates of *V. parahaemolyticus* as the environment continues to warm.

General Comments:

The manuscript was well-written. The arguments were easy to follow, well-reasoned, and supported by the data. I have relatively minor comments, mostly focusing on improving the quality of the figures and tables.

Minor comments:

- Some further grammatical editing is needed
- Write out all abbreviations at first appearance in the manuscript

Abstract

- line 32: Write out ST here

Introduction

- line 73: Resolve the referencing issue

Methods

- line 324: "Trimmomatic was used ..."
- line 325: "HiSeq"
- line 327: "Prokka was used ..."
- line 345: Reword this sentence fragment to improve clarity
- line 349: Reword this sentence to improve clarity
- line 377: "was completed using SWISS-MODEL"

Results

- Figure 3 should probably be in supplemental material
- Figure 4 is difficult to read as the panels are dense with small font text. Consider placing panel A above panel B so that both panels can be larger in the published document.
- Figures 5 and 6 need a title for the y-axis. Also, consider merging these two figures into one with an A and B panel.
- For Figure 7, consider placing panel A above panel B so that both panels can be larger in the published document.
- Consider merging Figures 8 and 9 into one Figure with an A and B panel.
- Tables 1 and 2 are far too large to be included in the manuscript body. They need to go into supplemental material.
- Change the orientation of the current Table 3 to vertical so that it can all fit on a single page in the body of the manuscript. Also, some of the white space can be removed.

- line 136: change "were analyzed" to "was used to analyze"
- line 149: This information is not presented in Figure 3. How was it determined that isolates were dispersed and non-regionally, and no host was related?
- line 200: Reference Figure 6 here?
- lines 202-204: Include references for these statements
- line 681: COGs were defined in the first sentence of this legend for Fig 2

Discussion

- line 282: It is unclear what the end of this sentence means
- line 295: "mutated" rather than "mutant"?

Staff Comments:

Preparing Revision Guidelines

Please return the manuscript within 60 days; if you cannot complete the modification within this time period, please contact me. If you do not wish to modify the manuscript and prefer to submit it to another journal, please notify me of your decision immediately so that the manuscript may be formally withdrawn from consideration by Microbiology Spectrum.

Review comments on “The impact of global warming on the signature virulence gene thermolabile hemolysin gene of *Vibrio parahaemolyticus*”

The methods of the paper are robust, and the tie-in to global warming is prescient. Minor misspelling or grammar corrections are required, as well as clarifications on methods or results.

Line 68: Change “aggregated” to “aggravated”.

Line 89: Clarify why risk of infection is indirect?

Line 197: Remove italics from “strains”

Line 226-228: Clarify if other unlabeled high-frequency mutation hotspots in Figure 10 not chosen for further analysis were initially considered, but had a protein-protein binding probability of less than 50%.

Line 324: We used, or, Trimmomatic was used

Line 327: We used Prokka, or, Prokka was used.

Line 295: Remove space between Fig, to stay consistent in the paper

Line 673: Typo, change to Figure Legend.

S1: Does this analysis take an increase in surveillance or strain collection into account?

Manuscript ID: Spectrum01502-23

Institute of Microbiology, Chinese Academy of Sciences
NO.1 Beichen West Road, Chaoyang District, Beijing 100101, China
CAS Key Laboratory of Pathogen Microbiology and Immunology

Kaixia Mi, Ph.D.

E-mail: mik@im.ac.cn

2023-08-16

Dear Dr. Theis,

We appreciate the reviewers for their positive response and interest in our work. We are sincerely grateful to the reviewers for their invaluable feedback, which helps us improve our manuscript entitled "*The impact of global warming on the signature virulence gene thermolabile hemolysin gene of Vibrio parahaemolyticus*". In response to the reviewers' comments, the manuscript, in its current form, has been edited and modified extensively and proof-read.

Our point-by-point response to the comments is given below.

Response to reviewers

Response to reviewer #1:

The methods of the paper are robust, and the tie-in to global warming is prescient. Minor misspelling or grammar corrections are required, as well as clarifications on methods or results.

1. Line 68: Change "aggregated" to "aggravated".

Thank you for the correction. The word is changed as suggested (Page 4, line 68 of the made up manuscript (pdf version)).

2. Line 89: Clarify why risk of infection is indirect?

Thank you for pointing out the misunderstanding. We would like to clarify that our intention was to highlight the fact that while *Vibrio parahaemolyticus* thrives in warmer waters, there have been reports of outbreaks occurring in colder regions, such as Alaska.

It appears that rising temperatures have been implicated in one of the largest known *V. parahaemolyticus* outbreaks. However, it is important to note that temperature alone does not directly cause infection or result in clinical symptoms. We have revised the manuscript to clarify our point as follows: “Despite *V. parahaemolyticus* thriving in warmer waters, outbreaks of *V. parahaemolyticus* have been reported in colder regions, including Alaska, where rising temperature plays an important role in contributing to these outbreaks.” (Page 5, lines 87-90 of the maked up manuscript (pdf version)). To enhance accuracy, we have also updated the reference in question. Reference 12 now provides more precise information: “McLaughlin JB, DePaola A, Bopp CA, Martinek KA, Napolilli NP, Allison CG, Murray SL, Thompson EC, Bird MM, Middaugh JP. 2005. Outbreak of *Vibrio parahaemolyticus* gastroenteritis associated with Alaskan oysters. *N Engl J Med.* 353(14):1463-70. <https://doi.org/10.1056/NEJMoa051594>.” (Reference 12 in the revised manuscript).

3. Line 197: Remove italics from "strains"

Thank you for pointing out the error. The word is modified as suggested (Page 10, line 199 of the maked up manuscript (pdf version)).

4. Line 226-228: Clarify if other unlabeled high-frequency mutation hotspots in Figure 10 not chosen for further analysis were initially considered, but had a protein-protein binding probability of less than 50%.

Thank you for pointing out the misunderstanding. Initially, we identified the mutation hotspots with high frequency and subsequently conducted additional analyses. To provide further clarity, we made the following modifications: “We performed a comprehensive analysis of all *tlh* full-length sequences (1,257 bp), including 778 clinical strains, 739 environmental strains, and 241 strains collected in our study. These sequences were compared with the *tlh* gene of the reference strain (RIMD2210633). Through this analysis, we identify a total of 48 SNPs. We calculated the frequency of mutations at each site and determined sites with a mutation frequency higher than 50% as high-frequency mutation points. We found 7 high-frequency mutation hotspots:

A180G, T552G, G657T, T858C, C1062T, A1137G, and T1179C (Fig.7). Additionally, we identified 2 specific sites, T259C and A951T, which were exclusively present in the clinical isolate (highlighted in red in Fig.7). These mutations exhibited a high frequency of occurrence exceeding 15% among clinical strains. To further analyze the potential significance of these mutations, we calculated the protein-protein binding probability for the identified mutation sites (Table S5). Our analysis revealed substantial variation in the ability of different sites to engage in protein-protein interactions, with probabilities ranging from 0.8% to 59.3% (Table S5). Notably, 2 mutation sites were predicted to be associated with a high probability of protein-protein binding sites (PPBS) (Table S5). These findings suggest a potential link between these mutations and pathogenicity through protein-protein interactions. However, experimental validation is necessary to confirm these predictions.” (Pages 11-12, lines 225-242 of the maked up manuscript (pdf version)).

5. Line 324: We used, or, Trimmomatic was used

Thank you for your suggestion. We change the sentence into “Trimmomatic was used to intercept adapters from sequencing reads generated by Illumina Hiseq” as suggested (Pages 16-17, lines 340-341 of the maked up manuscript (pdf version)).

6. Line 327: We used Prokka, or, Prokka was used.

Thank you for your suggestion. We change the sentence into “We used Prokka (v1.13.3) to annotate the genome and Prodigal (v2.6.3) to identify open-reading frames” (Page 17, lines 343-344 of the maked up manuscript (pdf version)).

7. Line 295: Remove space between Fig, to stay consistent in the paper

Thank you for your suggestion. We remove the space between “Fig” as suggested. (Page 15, line 314 of the maked up manuscript (pdf version)).

8. Line 673: Typo, change to Figure Legend.

Thank you for the correction. The word is changed as suggested (Page 33, line 696 of

the maked up manuscript (pdf version)).

9. S1: Does this analysis take an increase in surveillance or strain collection into account?

Yes, our analysis takes into account an increase in surveillance and strain collection over time. We collected data from the NCBI database, which includes records from 1950 to 2020. We then examined the relationship between the number of SNPs in the *tlh* and temperature and visualized this relationship using a generalized linear model.

Response to reviewer #2:

The manuscript aims to provide further genomic understanding of *Vibrio parahaemolyticus* and to specifically develop understanding of diversity in its virulence factors and how the mutation potential of some of the key factors correlates with ambient temperatures.

The general approach of the study was to sequence the genomes of 241 strains that were isolated from aquatic products imported or exported from China Customs between 2005-2010. Specific emphasis was placed on variation in the thermolabile hemolysin gene (*tlh*) gene in the context of temperature.

The conclusion of the study was that the *tlh* gene is likely to be an important marker of the pathogenic potential of environmental isolates of *V. parahaemolyticus* as the environment continues to warm.

General Comments:

The manuscript was well-written. The arguments were easy to follow, well-reasoned, and supported by the data. I have relatively minor comments, mostly focusing on improving the quality of the figures and tables.

Minor comments:

- Some further grammatical editing is needed
- Write out all abbreviations at first appearance in the manuscript

We sincerely appreciate your support of our paper. We have addressed the grammar-

related issue and expanded the abbreviations by providing their full forms upon their first mention in the modified manuscript.

1. line 32: Write out ST here

Thanks for your suggestion. We change the sentence into “In this study, a total of 241 strains were isolated from aquatic products imported or exported through China Customs between 2005-2010. The whole genomes of those strains were sequenced, revealing a highly significant level of genetic diversity. Our analysis identified 27 new sequence types (STs) ranging from ST2950 to ST2976.” as suggested (Page 2, lines 30-33 of the made up manuscript (pdf version))

2. line 73: Resolve the referencing issue

Thanks for the correction, we have resolved it (Page 4, line 73 of the made up manuscript (pdf version))

3. line 324: "Trimmomatic was used ..."

Thank you for your suggestion. We change the sentence into “Trimmomatic was used to intercept adapters from sequencing reads generated by Illumina Hiseq” as suggested (Pages 16-17, lines 340-341 of the made up manuscript (pdf version)).

4. line 325: "HiSeq"

Thank you for the correction. We have made the correction, changing it to “Hiseq”, as suggested (Pages 16-17, lines 340-341 of the made up manuscript (pdf version))

5. line 327: "Prokka was used ..."

Thank you for your suggestion. We change the sentence into “We used Prokka (v1.13.3) to annotate the genome and Prodigal (v2.6.3) to identify open-reading frames” (Page 17, lines 343-344 of the made up manuscript (pdf version))

6. line 345: Reword this sentence fragment to improve clarity

Thank you for your suggestion. We change the sentence into “To gain insight into the clonal relationships among the analyzed *V. parahaemolyticus* isolates, we conducted an analysis of their STs using goeBURST (v1.2.1) available at <http://goeburst.phyloviz.net>. The analysis focused on identifying isolates belonging to the same clonal complex, which was determined by the presence of a minimum of five out of seven identical STs.” (Pages 17-18, lines 359-363 of the maked up manuscript (pdf version))

7. line 349: Reword this sentence to improve clarity

Thank you for your suggestion. We modify the sentence into “Whole-genome sequences (WGS) were obtained from 241 *V. parahaemolyticus* strains (Table S2), along with 3 additional strains (VppAsia RIMD221063, VppUS2 FDA-R31, and VppX 10329 representing the Asian region, American region, and Pacific coast, respectively). WGS-SNP analysis was performed using KSNP (v3.1) with a kmer length of 20 to analyze these sequences. The resulting data were utilized to construct maximum likelihood phylogenetic trees using iqtree (v2.0.3). Model Finder was employed with 1000 iterations to perform optimal model selection for this analysis.” (Page 18, lines 365-372 of the maked up manuscript (pdf version))

8. line 377: "was completed using SWISS-MODEL"

Thank you for pointing out the error. We change the sentence into “The prediction of *t1h* protein structure was completed using SWISS-MODEL (<https://swissmodel.expasy.org/>).” (Page 19, lines 399-400 of the maked up manuscript (pdf version))

9. Figure 3 should probably be in supplemental material

Thank you for your advice. We have included it in the supplemental material, named Figure S1 as suggested.

10. Figure 4 is difficult to read as the panels are dense with small font text. Consider placing panel A above panel B so that both panels can be larger in the

published document.

Thank you for your advice. We have made revisions to Figure 4 in the revised manuscript. In order to improve readability and enhance clarity, we have divided the previous figure into two distinct figures: Figure 3A and Figure 3B. This modification allows for larger panels in the published document, thereby providing a better visualized of the data.

11. Figures 5 and 6 need a title for the y-axis. Also, consider merging these two figures into one with an A and B panel.

Thank you for your advice. We have added titles to the y-axis in both Figures 5 and 6 to enhance clarity and comprehension. Furthermore, we have merged these two figures into a single figure, labeled as Figure 4 in the revised manuscript.

12. For Figure 7, consider placing panel A above panel B so that both panels can be larger in the published document.

Thank you for your suggestion. We have made modifications to Figure 7 in the revised manuscript. First, we have repositioned panel A above panel B, to improve the layout and allow both panels to occupy larger space. Additionally, as a result of these changes, we have renamed this figure as Figure 5 in the revised manuscript.

13. Consider merging Figures 8 and 9 into one Figure with an A and B panel.

Thank you for your suggestion. We have combined Figures 8 and 9 into a cohesive single figure labeled Figure 6 in the revised manuscript, as suggested. The merged figure now includes an A panel and a B panel, providing improved organization and presentation of the data.

14. Tables 1 and 2 are far too large to be included in the manuscript body. They need to go into supplemental material.

Thank you for your advice. In consideration of the size of Tables 1 and 2, we have relocated them to the supplemental material. They can now be found in Table S2 and

Table S4 of the the revised manuscript, respectively.

15. Change the orientation of the current Table 3 to vertical so that it can all fit on a single page in the body of the manuscript. Also, some of the white space can be removed.

Thank you for your suggestion. We made an effort to change the table orientation to a vertical layout. However, despite our attempts, we encountered difficulties in achieving the aim to fit the entire table on a single page in the body of the manuscript. Nevertheless, we have implemented alternative modifications to address this concern effectively:

1. In order to enhance clarity and efficiency, we have implemented abbreviations in the table header. Specifically, we explored abbreviations for transition (TS), transversion (TV), non-synonymous mutations (NSM), and synonymous mutations (SM).
2. To optimize space utilization, we have minimized unnecessary white space within the table, ensuring a more compact presentation.

We believe with these adjustments, we have effectively resolved the issue of accommodating the table within a single page in the revised manuscript. It can now be found in Table 1.

16. line 136: change "were analyzed" to "was used to analyze"

Thank you for pointing out the error. We change the sentence into “Gonzalez-Escalona et al. described a multilocus sequence typing (MLST) scheme for *V. parahaemolyticus*, which relies on 7 housekeeping loci (*recA*, *dnaE*, *gyrB*, *dtdS*, *pntA*, *pyrC*, and *tnaA*) spanning both chromosomes of *V. parahaemolyticus*. This protocol was used to analyze the 241 isolates screened in this study.” (Page 7, lines 133-136 of the maked up manuscript (pdf version)).

17. line 149: This information is not presented in Figure 3. How was it determined that isolates were dispersed and non-regionally, and no host was related?

We agree with your point regarding the absence of information on isolates dispersed

and host-relatedness. We have revised the sentence as follows: “The analysis of the strains collected in this study revealed a dispersed distribution without regional clustering.” (Page 7, lines 148-150 of the maked up manuscript (pdf version)).

18. line 200: Reference Figure 6 here?

Thank you for your comment. We have included the reference to the figure in question and made adjustments accordingly. As a result, the designated reference figure has been updated from Figure 6 to Fig.4B of the maked up manuscript (pdf version). (Page 10, line 202 of the maked up manuscript (pdf version)).

19. lines 202-204: Include references for these statements

Thank you for your comment. To address the identified issue, we have made the necessary correction and additionally included reference 29 of the maked up manuscript (pdf version) “Mora C, McKenzie T, Gaw IM, Dean JM, von Hammerstein H, Knudson TA, Setter RO, Smith CZ, Webster KM, Patz JA, Franklin EC. 2022. Over half of known human pathogenic diseases can be aggravated by climate change. *Nat Clim Chang.* 12(9):869-875. <https://doi.org/10.1038/s41558-022-01426-1>.” as requested (Page 10, line 206 of the maked up manuscript (pdf version)).

20. line 681: COGs were defined in the first sentence of this legend for Fig 2

Thank you for your advice. In response to your advice, we have removed the sentence “COGs indicate clusters of orthologous groups.” from the legend for Figure 2 of the maked up manuscript (pdf version). (Page 33, lines 700-703 of the maked up manuscript (pdf version)).

21. line 282: It is unclear what the end of this sentence means.

Thank you for your advice. We modify the sentence into “And as predicted by ScanNet (<http://bioinfo3d.cs.tau.ac.il/ScanNet/>), T259C and A951T may be particularly important for bacterial binding to the receptor since their protein-protein binding can be up to about 50%.” (Pages 14-15, lines 299-301 of the maked up manuscript (pdf

version)).

22. line 295: "mutated" rather than "mutant"?

Thank you for the correction. We have made the modification as suggested and updated the sentence as follows: “Consistent with previous studies, we showed that mutations occur in *tlh*, and *tlh* was progressively more mutated with increasing global temperature (Fig.6B, S2), which may be potentially harmful to humans.” (Page 15, lines 312-315 of the made up manuscript (pdf version)).

September 5, 2023

Dr. Kaixia Mi
Institute of Microbiology, Chinese Academy of Sciences
NO.1 Beichen West Road, Chaoyang District,
Beijing
China

Re: Spectrum01502-23R1 (The impact of global warming on the signature virulence gene thermolabile hemolysin gene of *Vibrio parahaemolyticus*)

Dear Dr. Kaixia Mi:

Your manuscript has been accepted, and I am forwarding it to the ASM Journals Department for publication. You will be notified when your proofs are ready to be viewed.

Sincerely,

Kevin Theis
Editor, Microbiology Spectrum
